# Endometrial Cancer Detection Using a Cervical DNA Methylation Assay (MPap) in Women with Abnormal Uterine Bleeding: A Multicenter Hospital-Based Validation Study

**DOI:** 10.3390/cancers14174343

**Published:** 2022-09-05

**Authors:** Kuo-Chang Wen, Rui-Lan Huang, Lin-Yu Chen, Tzu-I Wu, Chien-Hsing Lu, Tang-Yuan Chu, Yu-Che Ou, Chen-Hsuan Wu, Shih-Tien Hsu, Dah-Ching Ding, Ling-Hui Chu, Chien-Wen Chen, Heng-Cheng Chang, Yu-Shu Liu, Hui-Chen Wang, Yu-Chun Weng, Po-Hsuan Su, Hao Lin, Hung-Cheng Lai

**Affiliations:** 1Department of Obstetrics and Gynecology, School of Medicine, College of Medicine, Taipei Medical University, Taipei 110, Taiwan; 2Department of Obstetrics and Gynecology, Shuang Ho Hospital, Taipei Medical University, New Taipei City 235, Taiwan; 3Translational Epigenetics Center, Shuang Ho Hospital, Taipei Medical University, New Taipei City 235, Taiwan; 4Department of Obstetrics and Gynecology, Wan Fang Hospital, Taipei Medical University, Taipei 116, Taiwan; 5Department of Obstetrics, Gynecology, Taichung Veterans General Hospital, Taichung 40705, Taiwan; 6Department of Obstetrics and Gynecology, Buddhist Tzu Chi General Hospital, Tzu Chi University, Hualien 970, Taiwan; 7Institute of Medical Research, Tzu Chi University, Hualien 970, Taiwan; 8Department of Obstetrics and Gynecology, Kaohsiung Chang Gung Memorial Hospital, Chang Gung University College of Medicine, Kaohsiung 833, Taiwan; 9Guzip Biomarkers Corporation, Hsinchu County 302041, Taiwan or or; 10Phalanx Biotech, Hsinchu County 302041, Taiwan; 11Beijing USCI Medical Instrument Company Limited, Beijing 101104, China; 12Department of Obstetrics and Gynecology, School of Medicine, College of Medicine, National Defense Medical Center, Taipei 114, Taiwan

**Keywords:** abnormal uterine bleeding, early detection of endometrial cancer, DNA methylation biomarker, MPap, triage tool

## Abstract

**Simple Summary:**

We conducted a multicenter validation study using a methylation assay, named MPap, to detect EC. MPap is used to identify the DNA methylation status of two genes, *BHLHE22* and *CDO1*, from cervical scrapings, and the results are combined with age and body mass index. In two stages of validation, the sensitivity, specificity, and positive and negative predictive values were 92.5~92.9%, 71.5~73.8%, 39.8~40.2%, and 98.0~98.1%, respectively. The MPap test is a feasible alternative tool that provides physicians with a reference for assessing susceptibility to endometrial cancer.

**Abstract:**

Background: We describe a DNA methylation assay, named MPap test, using cervical scraping as an alternative technique for endometrial cancer detection. Methods: A multicenter hospital-based, two-stage validation study was conducted to validate the cancer detection performance of the MPap test. The MPap value was determined from the DNA methylation status of two genes (*BHLHE22, CDO1*) and combined with two other clinical variables (age, BMI). The cutoff threshold of the MPap value was established in stage 1 and validated in stage 2. A total of 592 women with abnormal uterine bleeding were enrolled from five medical centers throughout Taiwan. Results: In stage 1, the sensitivity, specificity, and positive and negative predictive values of the MPap test for detecting endometrial cancer were 92.9%, 71.5%, 39.8%, and 98.0%, respectively. These values were validated in stage 2, being 92.5%, 73.8%, 40.2%, and 98.1%. Moreover, MPap outperformed transvaginal ultrasound in sensitivity and negative predictive values for detecting endometrial cancer. When we applied the algorithm for triage of endometrial cancer detection by MPap in the Taiwan National Health Insurance dataset, we found that it may reduce invasive procedures by 69~73%. Conclusions: MPap may provide a feasible alternative for endometrial cancer detection and can be considered as a triage test to reduce unnecessary invasive procedures.

## 1. Introduction

After decades of efforts in cancer research, improvements in the incidence and survival of most cancers have been encouraging. However, one of the exceptions is endometrial cancer (EC), the most common gynecologic cancer, for which both the incidence and mortality rates are increasing [1]. According to GLOBOCAN 2020, approximately 417,000 cases of EC were reported, and nearly 97,000 deaths were caused by EC in 2020 [2]. The number of women with newly diagnosed EC is estimated to grow by 52.7% in 2040, reaching 544,178 total cases [3]. The mortality rate of EC is increasing, and the estimated average rate of this increase from 2018 to 2040 is predicted to be 16% every 5 years [4]. A trend showing a slightly increasing incidence of EC in younger women in the USA, New Zealand, and Taiwan (incidence rate, 2.29–6.89) has been observed. Up to 26% of patients are at a premenopausal age according to the annual report of Taiwan National Health Insurance (https://www.hpa.gov.tw/Pages/Detail.aspx?nodeid=269&pid=14913). The incidence rate of EC in young women with ethnic differences may be related to high body mass index (BMI) [5,6,7,8]. Early detection of EC has been a very important issue worldwide. So, new strategies to improve EC detection and treatment are urgently needed.

The main symptom of EC is abnormal uterine bleeding (AUB), which is a shared symptom among several gynecologic problems in women. Although AUB may alarm women, very few opt for clinical care due to the painful protocols and time-consuming process of diagnosis, with the underlying uncertainty of EC. Endometrial thickness (ET) measured using transvaginal ultrasound (TVUS) is recommended by the American College of Obstetricians and Gynecologists as an essential noninvasive procedure to evaluate the risk of EC. Although the sensitivity of this technique could be high if a low threshold is set for ET, its specificity is low [9]. This status makes the ET threshold for EC risk difficult to standardize and leads to a high false positive rate, which results in many unnecessary biopsies [10]. The most common biopsy technology used is dilatation and curettage with anesthesia. Although it is the gold standard for EC diagnosis, it is an invasive procedure (IVP) for patients. Furthermore, this method can lead to uterine perforation, infection, and hemorrhage. Although endometrial aspiration is a popular alternative for endometrial sampling (ES), it is often repeated to decrease the false negative rate, especially when there is no endoscopic guidance; therefore, targets can be easily missed [11]. Endometrial samples obtained using suction curettage in an outpatient setting may have a higher sensitivity and specificity than endometrial assessment using TVUS; however, the failure rate of this IVP can be ≤54% [12]. ES failure and insufficient sample collection via ES are not uncommon, especially in obese or postmenopausal women [13]. Therefore, the development of a molecular triage tool auxiliary to the existing IVP used to prescreen women at high risk of EC is urgently needed.

Studies on epigenetic silencing have revealed a role for DNA methylation in carcinogenesis [14]. DNA methylation may occur early in carcinogenesis and is sufficiently stable for analysis. The application of DNA methylation as a biomarker for cancer detection or patient stratification has been increasing. However, research on EC epigenomics, especially for screening purposes, is relatively limited. Our previous comprehensive methylomics study presented a methylation panel of *BHLHE22, CDO1,* and *CELF4* genes for predicting EC risk using cervical scrapings [15]. The proof-of-concept real-time polymerase chain reaction (PCR)-based detection of methylated *BHLHE22* and *CDO1* genes was further prototyped in a retrospective cohort with a sensitivity of 84.8% and a specificity of 88.0% for the diagnosis of EC [16]. Following previous studies, we designed an in vitro diagnostic (IVD) product, named MPap (methylation Pap), for EC detection, by combining two DNA methylation statuses (*BHLHE22* and *CDO1*) with two other clinical variables (age and BMI). This MPap has been approved by the Taiwan Food and Drug Administration (TFDA) to quantitatively detect methylation in the cervical scraping samples of women who have abnormal uterine bleeding by using qPCR technology.

In this multicenter hospital-based study, we aimed to validate the clinical predictive performance of MPap for EC and compare its performance with the same noninvasive that of TVUS using ET measurements.

## 2. Materials and Methods

### 2.1. Study Design

We conducted a multicenter hospital-based, two-stage validation study from February 2018 to July 2020. Women aged ≥40 years with AUB were enrolled from gynecological outpatient clinics. A gynecologist at the clinic or hospital collected a cervical sample using a conventional PAP smear and then submerged the sample in sterile phosphate-buffered saline. The procedure of cervical scraping was performed to obtain a sample of the cervix for testing. All specimens were collected according to institutional policy. The collected samples were prepared for genomic DNA extraction and bisulfite conversion. The bisulfite-converted DNA was analyzed by dual-color qPCR, using a CDO1 methylation probe (FAM) in combination with an internal control (COL2A1) probe (VIC), but in separation from an independent BHLHE22 methylation probe (FAM). The MPap test (methylation Pap), which used the DNA methylation status of *BHLHE22* and *CDO1* from the cervical scraping sample with GMP grade reagent, was conducted and combined with two other clinical variables (age and BMI). MPap findings were reported blindly without knowledge of pathology results. MPap, an in vitro diagnostic (IVD) test, has been approved by the Taiwan Food and Drug Administration (TFDA). Using qPCR technology, the MPap test quantitatively detects methylation biomarkers in the cervical scraping samples of women who have abnormal uterine bleeding. In stage 1 of this study, an optimal threshold of the MPap value was established. In stage 2, independent samples were examined against the stage 1 results. Furthermore, the performance of TVUS for detecting EC risk was compared head-to-head in terms of sensitivity, specificity, positive predictive value (PPV), and negative predictive value (NPV) with that of MPap. The study protocol and informed consent form were approved by the Institutional Review Boards (IRBs) of the Taichung Veterans General Hospital (SE18199B), Kaohsiung Chang Gung Medical Foundation (201800647B0C601), Hualien Tzu Chi Hospital, Buddhist Tzu Chi Medical Foundation (IRB107-65-B), and Taipei Medical University (Joint IRB N201712038).

### 2.2. Patients and Samples

An informed consent form was signed by every participant prior to data and specimen collection. A total of 592 women aged ≥40 years with AUB were assessed in gynecological outpatient clinics. The following exclusion criteria were used: (1) a history of gynecological or breast cancer or cancer therapy; (2) hysterectomy; (3) current pregnancy, postpartum status, or lactation; or (4) a cervical diagnosis of atypical squamous/glandular cells of uncertain significance or worse. Participants with incomplete clinical and pathology results were excluded from the data analysis. Hence, 494 participants were enrolled and eligible for the final analysis (Figure 1). A total of 249 participants were assigned to stage 1, and 245 to stage 2. According to the research of Dr. Hajian-Tilaki [17], we estimated the sample size for testing the sensitivity (or specificity) of a single diagnostic test with a power under 85%.

### 2.3. Data Collection

Patient information, including age, height, and weight, was collected during face-to-face interviews, and ETs were obtained using TVUS. Pathology parameters, such as histologic type, grade, and surgical stage, were obtained from participants’ hospital records. Endometrial specimens with common physiological changes, benign endometrial lesions, and precancerous endometrial lesions were used as the non-EC control group (Appendix A).

### 2.4. Specimen Processing (DNA Extraction, Bisulfite Conversion, and Quantitative Methylation-Specific PCR)

Cervical scrapings, as obtained for routine Pap smears, were obtained by gynecologists. Genomic DNA from all Pap smear samples was extracted using a QIAamp DNA Mini Kit (QIAGEN, Hilden, Germany) and stored at −20 °C. Bisulfite conversion was performed using 800 ng DNA and an EZ DNA Methylation Kit (D5008; Zymo Research, Irvine, CA, USA). Technicians who processed specimens were blinded to the clinical information. The methylation status of *BHLHE22* and *CDO1* was conducted by quantitative methylation-specific PCR (qMSP) with the MPap Methylation PCR Kit (MPap, Guzip Biomarkers Corp., Hsinchu County, Taiwan). In addition, a non-CpG region of the type II collagen gene (*COL2A1*) was used to normalize the amount of input bisulfited DNA. MPap Methylation PCR Kit is an in vitro diagnostic (IVD) test that has been approved by the Taiwan Food and Drug Administration (TFDA). The qMSP was performed using TaqMan technologies and a Rotor-Gene Q real-time PCR (QIAGEN) with the following programs: activation at 95 °C for 10 min, 50 cycles if denaturation at 95 °C for 10 s, annealing and extension at 60 °C for 40 s, and cooling at 40 °C for 45 s. For each qPCR run, positive and negative controls executed duplicated run, and the average Ct values were calculated. The qMSP was performed once for each specimen. The COL2A1 gene, a CpG island-free gene whose copy number was not affected by methylation status in the qMSP assay, was used as input reference. The reaction was considered invalid if the Ct value of COL2A1 was >35. In stage 1, all experimental procedures, including sample preparation, DNA extraction, bisulfite treatment, and quantitative real-time PCR, were performed in a lab at Shuang Ho Hospital. In stage 2, the clinical performance study, experiments were carried out by an independent accredited third-party ISO 15189 laboratory (https://www.curiemed.com.tw/) to avoid any unnecessary influence. 

### 2.5. The Frequency of Diagnosing EC among IVPs from the Taiwan Cancer Registry Database

To investigate the frequency of diagnosing EC among different IVPs, we collected data that were obtained from the Taiwan Cancer Registry and National Health Insurance Research Database. The Taiwan Cancer Registry is a population-based cancer registry established in 1979 that recruits hospitals with > 50-bed capacity throughout the country to report newly diagnosed malignant neoplasms. The National Health Insurance Research Database contains claims data for beneficiaries enrolled in the Taiwan National Health Insurance program, which is a compulsory single-payer system that has covered approximately 99% of the 23 million residents of Taiwan since 1995. The procedure codes for IVPs include dilatation and curettage (F80401C), endometrial biopsy (F55002C), diagnostic hysteroscopy (F28022C), hysteroscopic polypectomy (F80422B), hysteroscopic endometrial ablation (F80423B), and hysteroscopic myomectomy (F80415B). IVPs that were performed for women with a diagnosis of infertility (ICD-9-CM 628) were excluded. The frequency of diagnosing EC among IVPs was estimated as the number of EC cases diagnosed using IVP divided by the total number of IVPs performed.

### 2.6. Statistical Analysis

The prevalence of EC in women varies by country, race, and hospital level [18]. In Taiwan, primary care physicians refer patients with uncertain conditions to regional hospitals or medical centers for further management. Moreover, the prevalence rate of EC in women with AUB from referred centers was 16%. *BHLHE22* and *CDO1* methylation confers a sensitivity and specificity of approximately 90% for EC detection [15,16]. The sensitivity and specificity of TVUS were estimated to be approximately 70% and 60%, respectively. The sample size estimation was performed according to clinical diagnostic accuracy, sensitivity, and specificity, and it was based on a type-I error (α) of 0.05 and a type-II error (β) of 15%. The frequency or proportion of categorical variables and the mean and standard deviation of replicate variables were used to characterize the EC and non-EC groups. The chi-squared test (for proportions), *t*-test, or analysis of variance were used to analyze differences between or within groups. All significant differences were assessed using a two-tailed *p* < 0.05. Receiver operating characteristic curve analysis was used to obtain the area under the curve (AUC), sensitivity, specificity, PPV, and NPV with the 95% confidence interval (95% CI) as a point estimation. Confidence intervals (CI) were supported for ROC analysis by the bootstrapping procedure (bootstrapping 1000 times) [19]. MedCalc^®^ Statistical Software version 19 (MedCalc Software Ltd., Ostend, Belgium; https://www.medcalc.org; 2020) was used to establish the MPap cutoffs. The performance in correlation with EC was estimated using the 95% CI at 85% power. All data were analyzed using R statistical software for Windows (version 4.0.2, R Core Team, Vienna, Austria). In stage 2, the performance of MPap was further assessed using the bootstrap method combined with R software. Furthermore, we simulated noninvasive triaging by MPap before IVPs for all women indicated for IVPs in 2016 to demonstrate how an additional triage would impact the frequency of cancer diagnosis among different IVPs.

## 3. Results

### 3.1. Baseline Characteristics

The demographic characteristics of the participants are shown in Table 1. The impact of three clinical variables (age, BMI, ET) and two methylation statuses (*BHLHE22* and *CDO1*) were investigated in our study. Patients in the EC group were significantly older than those in the non-EC group (normal, benign, and precancerous; *p* < 0.0001). BMI was significantly higher among the patients with EC who were recruited in stage 2 (*p* = 0.0019). However, only in the stage 1, compared to the non-EC group, there was a statistical difference between the four groups in the endometrial thickness of EC. Moreover, the methylation status of *BHLHE22* and *CDO1* was also significant between EC and non-EC groups. No significant differences in cell type, stage, or histological grade of EC were observed for patients recruited in either stage (Appendix A).

### 3.2. DNA Methylation Levels and Diagnostic Performance

Regarding the evaluation and validation of MPap for EC detection, we calculated an MPap value by combining different variables including methylation status of *BHLHE22* and *CDO1*, age, and BMI in stage 1. We used the following formula (*β* being weight) to determine the MPap value.
MPap value=β_1_ BHLHE22 + β_2_ CDO1+ β_3_ Age +β_4_ BMI + ϵ
Where: ϵ = −5.18(1)

By combining these four factors, the AUC of the MPap value was 0.91 (0.87–0.94) (Table 2 and Figure 2A). 

Compared to the non-EC group, there was a statistically significant difference among the four groups in MPap values for EC (Appendix A). The optimal cut-off threshold of the MPap value for detecting EC was established in stage 1 (Figure 2B, dotted line). The MPap value above of −2.10 was considered to be high risk for EC, whereas below −2.10 can be considered low risk. The sensitivity and specificity of MPap were 92.9% (80.5–98.5%) and 71.5% (64.8–77.5%), respectively (Appendix A). The PPV and NPV were 39.8% (34.4–45.5%) and 98.0% (94.3–99.3%), respectively. These results were validated in stage 2, and the AUC was 0.90 (0.84–0.95) (Figure 2C). The sensitivity and specificity were 92.5% (82.9–100.0%) and 73.8% (67.6–79.4%), respectively. The PPV and NPV were 40.2% (30.8–50.5%) and 98.1% (95.8–100.0%), respectively (Appendix A). The optimal cut-off threshold of the MPap value for identifying EC, with a score setting at −2.10, was ideally validated in stage 2 (Figure 2D). In the comparison of the performance of MPap for participants from centers in different areas (northern, central, and southern Taiwan), there were no significant differences (*p* = 0.39) among centers (Appendix A). Except for histological type I (endometrioid), MPap can be used to detect type II EC (Appendix A). MPap could also detect rare types of EC, including carcinosarcoma, clear-cell carcinoma, and serous-type carcinoma.

### 3.3. MPap and Endometrial Thickness

In addition, we performed a comparison between MPap and TVUS. Both procedures can be carried out largely on an outpatient basis. TVUS uses different thicknesses as an ET threshold by AUC in the detection of EC (Appendix A). The ROC was calculated according to 233 patients with endometrial thickness (ET) in stage 1. The best performance of area under the curve (AUC) was 0.65 (95% CI: 0.55–0.75) for an ET of 16 mm for detecting EC. The sensitivity and specificity were 50.0~55.6% and 77.7~82.2% (71.2–83.3), respectively. During the head-to-head comparison, MPap outperformed TVUS ET in terms of the sensitivity and negative predictive value, and the results were statistically significant (Table 3).

### 3.4. MPap in Algorism

Based on the above finding, our MPap assay may be considered to be an alternative procedure for detecting EC clinically. Compared to TVUS, MPap has comparable benefits including being noninvasive and having a better sensitivity and NPV. In Taiwan, TVUS examination is available in most gynecological clinics. Tissue specimen proof for the diagnosis of EC using various IVPs is a standard of care (Figure 3A). Therefore, we set a clinical speculation of how many IVPs would be diminished if MPap were applied in existing databases, using the sensitivity (92.5%) and specificity (73.8) of MPap from our results in stage 2. This could estimate the impact of the MPap test in clinical practice. We proposed an algorithm with triage by MPap for the detection of EC in patients with AUB who are ≥40 years old (Figure 3B, Appendix A).

Lastly, we applied this algorithm to National Health Insurance data to simulate clinical status. According to the 2014 Taiwan National Health Insurance data (Appendix A), people over 50 years of age received a total of 17,752 IVPs, and an average of 15.9 cases of endometrial cancer were diagnosed. After triage with MPap, 5393 IVPs were required for high-risk patients, and an average of 5.2 cases of endometrial cancer were diagnosed. A total of 69.1% of IVPs (12,275/17,752) could be diverted. The 40~50-year-old group received a total of 20,951 IVPs, and an average of 86.6 cases of endometrial cancer were diagnosed. After MPap shunting, 5650 IVPs were required for high-risk patients, and an average of 25.2 cases of endometrial cancer were diagnosed and could be triaged. A total of 72.9% of IVPs (15,283/20,951) could be diverted. 

## 4. Discussion

This study demonstrates that the MPap test (methylation Pap), which uses the DNA methylation status of two genes, namely, *BHLHE22* and *CDO1*, together with age and BMI, can be a useful biomarker for the triage of women with AUB aged ≥40 years using cervical scraping. Both age and BMI are important clinical factors that may influence EC survival outcomes but are inadequate [20]. Additional markers to synergize the value of age and BMI are clinically urgent. The MPap test is an in vitro diagnostic (IVD) test that has been approved by the Taiwan Food and Drug Administration (TFDA). Using qPCR technology, the MPap test quantitatively detects methylation biomarkers in the cervical scraping samples of women who have abnormal uterine bleeding. The sensitivity of this test can be >90%, with a specificity of approximately 75%, which may substantially reduce unnecessary IVPs or referrals for IVPs. Using the National Health Insurance database simulation, this new algorithm may substantially reduce the use of IVPs by approximately 70%. The MPap test, as an auxiliary diagnostic tool or alternative method, provides a physician with a reference of a patient’s susceptibility to EC to justify the necessity of confirmatory diagnosis, if not therapeutic, using an IVP. The test results should be used in combination with a physician’s assessment and individual risk factors in guiding patient management. In other words, the MPap test is a pre-IVP screening aid or an alternative for EC risk mitigation.

Methylation biomarkers for EC detection have been reported using minimally invasive sampling, such as Tao brush sampling (intrauterine sampling), cervical scraping, vaginal tampon sampling, and self-collected brush device sampling [21,22,23,24,25]. Bakkum-Gamez et al. [21] also found hypermethylated *HTR1B*, *RASSF1*, and *HOXA9* genes in patients with EC using DNA from intravaginal tampons. A targeted sequencing test of somatic mutations in 18 genes and aneuploidy, PapSEEK, detected 81% of EC cases at a specificity of 99% using the Pap brush and at a sensitivity of 93% and specificity of 100% using the Tao brush in a case–control setting [23]. However, mutations were observed more frequently than expected in some studies. The study of uterine lavage samples by Maritschnegg et al. [26] showed that 29.6% of patients with a benign gynecological condition, including uterine myomas, ovarian cysts, or ovarian teratomas, harbored a genetic mutation. PTEN and TP53 mutations were commonly detected in the cervical scrapings of healthy controls and patients with benign gynecologic diseases. Clonal proliferation of nonmalignant cells and benign diseases has been described in bone marrow, noncancerous tissue, and endometriotic lesions [27,28]. Although these mutations might reflect benign or noncancerous endometriotic lesions, the mechanism underlying the mutational changes in normal endometrium and benign uterine lesions remains to be elucidated [28,29]. Further validations are needed to test the feasibility of genetic mutations in clinical practice. The hypermethylated *BHLHE22* and *CDO1* genes, which are detectable in cervical scrapings, have been tested in a retrospective cohort in the discovery and verification phases with high sensitivity and specificity [16]. In this multicenter validation setting, we validated MPap as a useful strategy for the triage of women with AUB for the detection of EC.

The role of TVUS in EC screening remains controversial. There is no survival advantage of screening asymptomatic women using TVUS or ES for EC [30]. Although TVUS is considered to be good at detecting EC in women with postmenopausal bleeding (PMB) with high sensitivity and specificity at an ET cutoff of >4 mm, its discretion power for symptomatic premenopausal women is limited [31,32]. Clinically, we still only have TVUS to apply as a noninvasive procedure for AUB now. So, we compare the status of TVUS to MPap, which is also considered a noninvasive procedure. The PPV in the present study of women aged ≥40 years with AUB was very low (approximately 15%) at an ET cutoff of 5 mm and not much higher at a cutoff of ≤16 mm (approximately 20%), resulting in a rather high number of false positives. Thus, TVUS generates excessive IVP referrals for biopsies. Despite the availability of more convenient aspiration devices than dilatation and curettage, the latter is used up front to evaluate women with AUB. Moreover, 20–36% of the negative results of dilatation and curettage are because of sampling failures, especially at corners of the endometrial cavity [33]. A recent meta-analysis of 40,790 women reported that only 9% (95% CI 0.08–0.11) of women with PMB are diagnosed with EC [18]. Therefore, a useful triage tool for EC risk prediction is urgently needed to expedite and improve women’s health decision-making. As the participant hospitals are referred centers, the EC prevalence among women with AUB in these hospitals was higher than that in the community populations. Further population studies in different geographic areas are needed to test the impact of MPap in the real-world setting. The best diagnostic strategy for patients suspected of having EC remains controversial. Currently, identification of abnormalities using TVUS followed by endometrial biopsy is a widely accepted strategy for cases where EC is suspected. However, critical findings are often missed by novice TVUS operators. Such missed findings indicate the requirement of experience to operate and interpret TVUS. In the UK and Canada, primary care professionals unfamiliar with TVUS often encounter patients with AUB, which is a common gynecologic condition for which the patient needs to be referred to a gynecology specialist within 2 weeks [25]. To alleviate this problem, MPap as a triage for women with AUB who are at high risk for EC may help primary care professionals make rapid decisions regarding patient referral. Such integration of MPap into EC diagnostics is noninvasive as TVUS and reliable for women aged ≥40 years with AUB. 

This is the first study to validate the utility of MPap using cervical scrapings with the least physical injury in routine clinical practice. Moreover, the performance of MPap indicates a reliable triage method for EC diagnosis in AUB cases to avoid unnecessary IVPs for women at low risk of EC. The strength of the present study is the use of a standardized DNA methylation assay and its validation in multiple clinical settings with sufficient statistical power. Limitations of this study include its hospital-based design and results that are not applicable to the general population, especially asymptomatic women or those < 40 years old. The MPap performance for women, who are at high risk for EC such as those with Lynch syndrome or undergoing tamoxifen treatment, may be later investigated. Finally, further longitudinal study for those with positive MPap results but no EC are needed to clarify the predication value of MPap. 

## 5. Conclusions

In conclusion, the MPap test was promising as an alternative for EC detection in women aged ≥40 years with AUB.

## Figures and Tables

**Figure 1 cancers-14-04343-f001:**
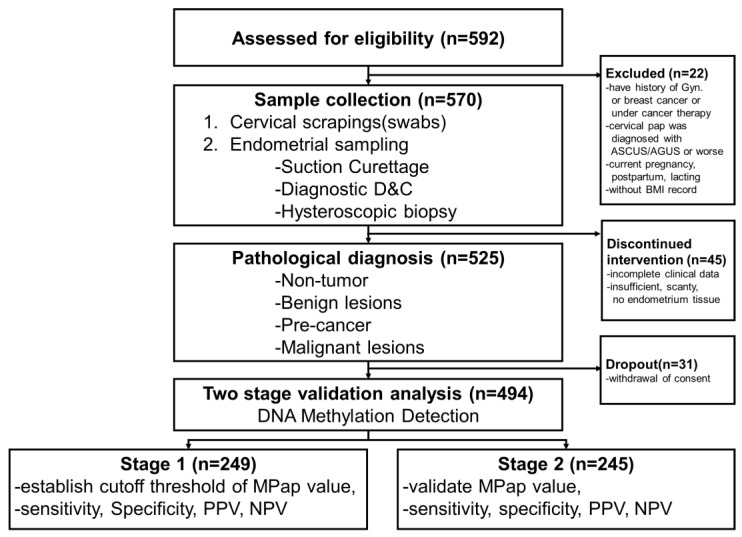
Trial profile.

**Figure 2 cancers-14-04343-f002:**
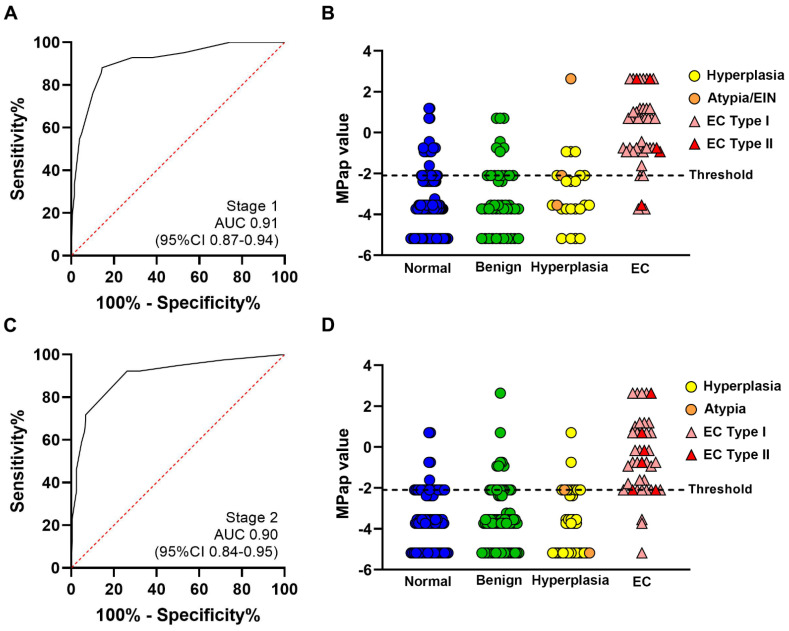
Performance of MPap for EC detection. All results are from eligible subjects. Dots and triangles represent MPap; (**A**) result of receiver operating characteristic curve analysis of stage 1, AUC = 0.91; (**B**) MPap result comparison of the four pathology groups in stage 1; (**C**) results of receiver operating characteristic curve analysis in stage 2, AUC = 0.90; (**D**) MPap result comparison of the four pathology groups in stage 2.

**Figure 3 cancers-14-04343-f003:**
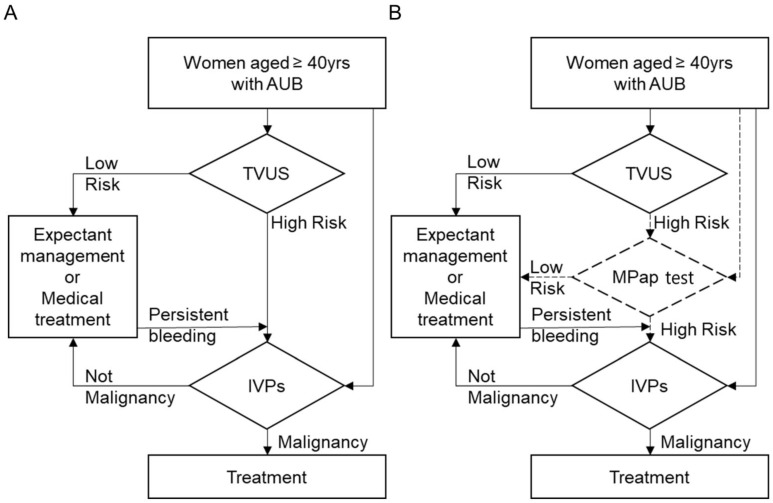
Diagnostic flowchart for EC in clinical practice. (**A**) The current diagnostic procedure; (**B**) the triage suggested using MPap. AUB, abnormal uterine bleeding; TVUS, transvaginal ultrasound; IVPs, invasive procedures.

**Table 1 cancers-14-04343-t001:** Demographic characteristics of the participants.

Cohort		Normal	Benign	Precancerous	EC	*p*-Value
Stage 1	n	138	45	24	42	
	Age, mean (SD)	49.3 (7.2)	51.0 (9.2)	48.0 (6.2)	56.7 (8.5)	<0.0001
	BMI, mean (SD)	24.0 (4.2)	25.1 (5.4)	25.6 (5.8)	25.7 (4.8)	0.0966
	ET, mean (SD)	11.6 (9.2)	13.8 (6.9)	11.8 (7.0)	16.4 (8.8)	0.0223
	BHLHE22 (Ct)	49.7 (1.8)	49.8 (1.4)	49.6 (2.2)	45.6 (6.4)	<0.0001
	CDO1 (Ct)	46.0 (5.0)	46.2 (4.2)	45.2 (5.4)	38.4 (5.1)	<0.0001
Stage 2	n	86	83	37	39	
	Age, mean (SD)	49.4 (6.7)	49.0 (6.9)	48.3 (5.0)	59.9 (11.0)	<0.0001
	BMI, mean (SD)	24.1 (4.6)	24.0 (3.7)	24.1 (3.6)	27.1 (6.2)	0.0019
	ET, mean (SD)	11.4 (13.6)	13.8 (18.6)	11.7 (4.7)	16.4 (8.1)	0.2957
	BHLHE22 (Ct)	50.0 (0.0)	49.9 (1.4)	50.0 (0.0)	46.1 (6.2)	<0.0001
	CDO1 (Ct)	48.0 (3.4)	47.6 (4.1)	47.8 (3.6)	42.4 (6.2)	<0.0001

Three clinical variables: Age, years; BMI, body mass index (kg/m^2^); ET, endometrium thickness (mm).Two methylation variables: *BHLHE22* and *CDO1*, cycle threshold value (Ct).

**Table 2 cancers-14-04343-t002:** Composition of different combined variables from the cases in stage 1.

Composition of Different Combined Variables	AUC (95% CI)
BHLHE22 + CDO1	0.86 (0.82 to 0.90)
BHLHE22 + CDO1 + Age	0.89 (0.84 to 0.92)
BHLHE22 + CDO1 + BMI	0.88 (0.83 to 0.92)
BHLHE22 + CDO1 + Age + BMI	0.91 (0.87 to 0.94)

AUC, area under the curve. CI, confidence interval. BMI, body mass index (kg/m^2^).

**Table 3 cancers-14-04343-t003:** Comparison of MPap and TVUS for EC detection.

		MPap-Value	TVUS	*p*-Value
		≥Cutoff Threshold	ET	
Stage 1	Sen	92.9 (80.5–98.5)	50.0 (32.9–67.1)	<0.0001
	Spe	71.5 (64.8–77.5)	77.7 (71.2–83.3)	0.1190
	PPV	39.8 (34.4–45.5)	29.0 (21.1–38.3)	0.0128
	NPV	98.0 (94.3–99.3)	89.5 (85.9–92.2)	0.0001
Stage 2	Sen	92.5 (82.9–100.0)	55.6 (38.1–72.1)	<0.0001
	Spe	73.8 (67.6–79.4)	82.2 (76.2–87.3)	0.0271
	PPV	40.2 (30.8–50.5)	36.4 (27.3–46.5)	0.3936
	NPV	98.1 (95.8–100.0)	91.0 (87.5–93.6)	0.0006

Data % are within the 95% CIs unless otherwise indicated. The cutoff threshold of MPap value is ‒2.10. Endometrium thickness of TVUS ≥ 16 mm is cutoff threshold. Sample size of MPap test: n = 249 in Stage 1; n = 245 in Stage 2. Sample size of TVUS: n = 233 in both stages.

## Data Availability

All data may be shared, subject to our institute’s data sharing policy, which has been approved by the Institutional Review Boards (IRBs). Data can be made available upon reasonable request. Requests should be made to the corresponding author.

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
