# Peer review of "Endometrial Cancer Detection Using a Cervical DNA Methylation Assay (MPap) in Women with Abnormal Uterine Bleeding: A Multicenter Hospital-Based Validation Study"

_cancers, 2022, doi:10.3390/cancers14174343_

Round 1

Reviewer 1 Report

Liew et al. describe the training and validation of a commercial methylation assay MPap for detection of Endometrial cancer in cervical scrapes of women with abnormal uterine bleeding. Compared to the current used reference transvaginal ultrasound (TVUS) the MPap assay displays a higher sensitivity at a lower specificity for endometrial cancer. This is a promising study and here are my comments to help the authors improve the manuscript.

1)    Please describe how patients were divided into the 2 study groups (e.g. randomization or selection).

2)    The value of only age and BMI without the methylation markers must be added to the manuscript as reference. Detection of EC in this high-risk population seems also to depend on these factors given their significant difference between the groups indicated in table 1.

This helps the reader to understand the true additive value of the methylation markers on top of age and BMI.

3)    The Mpap scores several EC as negative. The authors do not discuss why these cases are negative? Please provide descriptive information about cancer stage, age, bmi, bleeding, tvus score for the Mpap-negative cancer cases and address in the discussion section. This also helps to interpret the risk of the false-negative cancer cases in clinical practice (figure 3B).

4)    It is not clear from the manuscript and supplementary files how the methylation status of the genes BHLHE22 and CDO1 are calculated, is this only the Ct value of the gene or is it corrected for the Ct value of the COL2A1 housekeeping gene? Please add to the manuscript.

5)    Was testing of the MPap assay performed at the local institutes or was this performed in a central lab?

6)    The average Ct values of the BHLHE22 and CDO1 genes in the EC groups (table 1) are still very high with Ct values above 35, which generally reflect the threshold for very low target copy numbers (below 1-2 copies). Such low concentrations are very difficult to reproduce and display a large testing variety. Because duplicate measurements were performed, please indicate the coefficient of variation of the Ct of each gene.

For samples in which one of the duplicate measurement has a Ct value and the other is negative, how are the BHLHE22 and CDO1 values calculated?

7)    Please include reference to reproducibility data of the Mpap test preferably of clinical samples.

8)    Why is in the algorithm (figure 3B) the Mpap test only performed on high-risk TVUS, when it was demonstrated that mpap has a higher sensitivity than tvus?

9)    Change “detection of EC in patient with AUB” (line 293) to “detection of EC in patient with AUB and positive for TVUS”.

10)     The data of the women from 2014 used in the simulation, have all women been referred for IVP because of TVUS-positive result? Otherwise this needs to be aligned with the algoritm (fig 3B).

11)     How is the cut-off value of the MPap (-2.01) determined?

Author Response

Dear Reviewer 1:

Reviewer 2 Report

Endometrial Cancer Detection Using a Cervical DNA Methylation Assay (MPap) in Women with Abnormal Uterine Bleeding: A Multicenter Hospital-Based, Validation Study

Wen et al. (Cancers)

The authors conducted a multicenter study to validate the performance of the commercialized MPap test (targeting DNA methylation of the BHLHE22 and CDO1 genes) for endometrial cancer detection in physician-collected cervical scrapes of women >40y with abnormal blood loss. The total study population (n=494) was split into a training and test set to first establish a cutoff threshold of the MPap value and then validate this threshold using the remaining samples. Methylation levels were combined with clinical characteristics, including age and BMI, to estimate the risk for endometrial cancer. The accuracy for endometrial cancer detection by MPap was compared to transvaginal ultrasound and showed a higher sensitivity and specificity.

Major comments

1.       The rationale for focusing on BHLHE22 and CDO1 only is currently unclear. CELF4, HAND2, and TBX5 were also discovered as valuable biomarkers for endometrial cancer detection in cervical scrapings in the previous discovery study (Huang et al. 2017). These markers were also validated in a follow-up study in which the value of combining genetic and epigenetic biomarkers was assessed (Liew et al. 2019). Can the authors explain why the MPap test panel only contains BHLHE22 and CDO1, and not CELF4, HAND2, and/or TBX5?

2.       A dual-color qPCR was run for CDO1 and COL2A1 combined, with COL2A1 as a reference gene. Methylation levels of BHLHE22 were run in a separate reaction. Could the authors explain in more detail how the data was normalized in the BHLHE22 runs to ensure sufficient sample input? Especially because BHLHE22 Ct values were always 50 (undetermined) in the normal and precancerous groups tested in Stage 2 as presented in Table 1.

3.       The assignment of participants to stage 1 and stage 2 of the trial (training and test set) is not described in detail but of high importance. Could the authors describe further how these groups are formed? Was this random or based on particular clinical characteristics besides final diagnosis? The distribution of different pathological diagnoses are not equal between the training (stage 1) and test set (stage 2). Stage 2 contains less normal and more benign and precancerous cases as compared to stage 1. What’s the reason for this? The sentences 203 and 204 on the characterization of EC and non-EC groups are currently unclear.

4.       The study population was separated into two groups to validate the MPap threshold and assess the performance of the MPap test in an independent group. Apart from this validation, confidence intervals of the ROC curve were validated by bootstrapping. It is important for the reader to see whether the outcomes of this random resampling method yielded similar performances (AUC value, sensitivity and specificity).

5.       The significant difference in age between women without EC and women diagnosed with EC should be addressed in the limitations section of the discussion. Methylation levels increase with age and women with EC were significantly older, which might have led to an overestimation of the MPap test performance. The performance may be lower when tested in another population including age matched controls.

6.       It would be interesting to assess and describe the diagnostic value of Age + BMI only. This would provide more insight into the added value of the methylation markers tested within the MPap. Additionally, it would be interesting to examine whether TVU outcome could also complement the MPap assay, like age and BMI. As this test will supplement the assessment by a physician, both the individual risk factors and the TVU score will be available.

7.       The optimal cut-off used to classify cases as cancers was set at a MPap value of -2,01 within stage 1 of the study. The reasoning behind the definition of this particular threshold is not described. This information should be added as this is the main goal of stage 1 of this study and important for the performance outcomes of stage 2. Table S4, named ‘comparison of different thresholds of the MPap value for EC detection’, does not contain performance measures at different thresholds. It would be of interest to the readership to elaborate further on why this threshold was considered optimal and show performance measures of different thresholds in Table S4.

8.       The proposed flowchart (Figure 3, panel b) in which MPap is added to the diagnostic pathway, is not supported by data from this article. As stated in point 6, it would be of utmost value to a) calculate the sens and spec of TVUS+MPap before incorporating this into a diagnostic flowchart and b)calculate how the test performance of Mpap would be if preceded by a TVUS

Minor comments

1.       The English language could be improved at some points throughout the manuscript. Unnecessary text might be removed and grammatical errors should be corrected. For example, ‘The’ can be removed in sentence 63: ‘The After decades of efforts in cancer research, improvements in the incidence and 63 survival of most cancers have been encouraging.’, and also in Figure 1 and Figure 2 ‘The Stage-I and The Stage-2’. An example of a grammatical error is seen in line 313: ‘that has approved’ à that has been approved. Another example: 2.2. Patients and Ssamples or “ In addition, we performed a comparison between MPap and TVUS that both may be clinically available non-invasive procedure at outpatient departments.

2.       GLOBOCAN 2018 metrics on endometrial cancer incidence and mortality are used. Consider referring to the more recent GLOBOCAN Cancer Statistics of 2020.

3.       The statement that “ The most common biopsy technology used is dilatation and curettage with anesthesia. Although it is the gold standard for EC diagnosis, it is an invasive proce-88 dure (IVP) for patients.” Does not reflect common practice in the Western world where mimimal invasive techniques are widely used. More details: https://www.uptodate.com/contents/endometrial-sampling-procedures

4.       The use of mutational changes as biomarkers for endometrial cancer are widely discussed in the discussion section. Likewise, it would be of interest to elaborate further on other methylation markers described in literature and the performance of the MPap test as compared to other methylation markers and methylation marker panels.

5.       Methylation markers have also been described in the context of endometrial cancer detection in non-invasive sampling types using circulating tumor DNA, including plasma and urine (e.g. Beinse et al. 2022 and van den Helder et al. 2021). The discussion section would be strengthened when adding this to line 327.

6.       The use of the abbreviation IVP for invasive procedures is uncommon and confusing as this is also short for intravenous pyelogram (imaging of the urinary tract).

7.       Table S3 would be more clear if the authors describe how the MPap values are presented. It should be easier to understand whether these are average or median values and what is indicated by the numbers between brackets.

Author Response

Dear Reviewer 2:

Round 2

Reviewer 1 Report

The authors response sufficiently addressed the raised questions.